# RHS-elements function as type II toxin-antitoxin modules that regulate intra-macrophage replication of *Salmonella* Typhimurium

**Magnus Stårsta**[☯], **Disa L. Hammarlöf**[☯], **Marcus Wäneskog**, **Susan Schlegel**, **Feifei Xu**, **Arvid Heden Gynnå**, **Malin Borg**, **Sten Herschend, Sanna Koskiniemi** *

Department of Cell and Molecular Biology, Uppsala University, Uppsala, Sweden

☯ These authors contributed equally to this work.
* sanna.koskiniemi@icm.uu.se

**Data Availability Statement:** The data underlying the results presented in the study are available in S1 Data.

## Abstract

RHS elements are components of conserved toxin-delivery systems, wide-spread within the bacterial kingdom and some of the most positively selected genes known. However, very little is known about how Rhs toxins affect bacterial biology. *Salmonella* Typhimurium contains a full-length *rhs* gene and an adjacent orphan *rhs* gene, which lacks the conserved delivery part of the Rhs protein. Here we show that, in addition to the conventional delivery, Rhs toxin-antitoxin pairs encode for functional type-II toxin-antitoxin (TA) loci that regulate *S.* Typhimurium proliferation within macrophages. Mutant *S.* Typhimurium cells lacking both Rhs toxins proliferate 2-times better within macrophages, mainly because of an increased growth rate. Thus, in addition to providing strong positive selection for the *rhs* loci under conditions when there is little or no toxin delivery, internal expression of the toxin-antitoxin system regulates growth in the stressful environment found inside macrophages.

## Author summary

Bacteria that reside and multiply inside of phagocytic cells are hard to treat with common antibiotics, partly because subpopulations of bacteria are non-growing. Very little is known about how bacteria regulate their growth in the phagocytic vesicle. We show that RHS elements, previously known to function as mobilizable toxins that inhibit growth of neighboring bacteria, also function as internally expressed toxin-antitoxin systems that regulate *Salmonella* Typhimurium growth in macrophages. RHS elements were discovered more than 30 years ago, but their role in biology has long remained unclear even though they are some of the most positively selected genes known. Our results suggest an explanation to why *rhs* genes are under such strong positive selection in addition to suggesting a novel function for these toxins in regulating bacterial growth.

**Funding:** This study was supported by grants from the Swedish Foundation of Strategic Research (ICA12-0025) (https://strategiska.se/), the Swedish research council (E0239301)(https://www.vr.se/), the European research council (ERC-2018-STG-804068)(https://erc.europa.eu/) and the Wenner-Gren foundations (https://www.swgc.org/) to S.K. The funders had no role in study design, data collection and analysis, decision to publish, or preparation of the manuscript.

**Competing interests:** The authors have declared that no competing interests exist.

## Introduction

How bacteria regulate their growth during infection is of fundamental interest for bacterial physiology and for development of new improved treatment regimens for bacterial infections. Previous studies show that toxin-antitoxin (TA) modules are important for regulating the growth of bacteria within phagocytic vacuoles in immune cells [1, 2]. TA-systems are divided into six classes (I-VI) depending on the nature of the toxin and antitoxin (protein or RNA) and the way the antitoxin mediates protection (binding, degradation or regulation of expression) (reviewed in [3]). The toxin and antitoxin of type II TA-systems are both proteins and the antitoxin protect against toxicity of the cognate toxin by binding and blocking its activity. The antitoxins of type II TA-systems are actively targeted by cellular proteases and therefore less stable, i.e. have a shorter protein half-live than their cognate toxins [4, 5]. Degradation of antitoxins results in free unbound toxins that are able to corrupt essential cellular processes, which ultimately results in growth arrest [6, 7]. TA-modules were initially considered to function as "addiction modules" or "selfish-genetic elements" found on low-copy-number plasmids, where they provide plasmid stability through post-segregational distortion [8–10]. Daughter cells that lose the plasmid stop proliferating because the unstable antitoxin needs to be continuously produced from the plasmid to prevent self-intoxication by the more stable toxin [8–10]. Type II TA-systems were later also found to be abundant in bacterial genomes and have since been linked to bacterial virulence in many bacteria, including *Escherichia coli* [11], *Mycobacterium tuberculosis* [12], *Staphylococcus aureus* [13] and *Salmonella enterica* [14].

*Salmonella enterica* serovar Typhimurium (hereafter *S.* Typhimurium) is a facultative intracellular pathogen that causes mild self-limiting gastro-intestinal disease in humans [15] but also typhoid-like disease in many hosts including humans [15, 16]. *S.* Typhimurium has a number of virulence factors allowing it to survive and replicate within phagocytic cells and to cause systemic disease [17]. Previous findings suggest that *S.* Typhimurium cells enter a non-growing state upon entry to host macrophages, through a mechanism dependent on type-II TA-modules found in the *S.* Typhimurium chromosome [1].

Bacterial contact-dependent growth inhibition (CDI) systems comprise functional toxin-antitoxin modules, but where typical TA-modules are expressed inside the bacterial cell, the toxins of CDI systems are delivered between cells leaving the antitoxin inside the CDI producing cell. Rhs proteins are widely distributed CDI toxins, common throughout β-, γ- and δ-proteobacteria and with distantly related YD-peptide repeat proteins found also in Gram-positive bacteria and in higher vertebrates [18, 19]. In Gram-negative bacteria, Rhs toxins are secreted and mobilized by the type VI secretion system (T6SS) [20–22], which forms a needle-like cell-puncturing device that penetrates the membrane of the target cell and can deliver multiple effectors to both prokaryotic and eukaryotic cells [23, 24]. Effectors can be delivered through the needle-tube or by attachment to the tip of the puncturing device. The tip of the T6S devise is formed by e.g. the PAAR-domain containing Rhs toxins [25, 26]. Rhs proteins contain a delivery part, which folds into a cone-like structure forming a shell that encapsulates the highly variable C-terminal toxin domains demarcated by the PxxxxDPxGL motif [27, 28]. These C-terminal toxins are delivered to target cells upon cell-cell contact [20]. To protect themselves from auto-inhibition, bacteria with Rhs toxins also encode small, highly specific antitoxins or immunity proteins, that bind and block the toxic activity of their cognate toxins [20]. The activities of the various C-terminal domains range from nucleases to ADP-ribosylating toxins and many of the toxic activities remain unknown [20, 29–31].

*S.* Typhimurium contains a single *rhs* locus with a full-length *rhs* gene and one adjacent orphan toxin, which lacks the Rhs shell, hereafter referred to as the delivery part of the Rhs

protein (Fig 1A). The *rhs* locus is found adjacent to the genes encoding a T6SS on *Salmonella* Pathogenicity Island 6 (SPI-6) [32, 33]. In addition to their role in antagonistic interactions [20, 34], Rhs proteins have been shown to modulate host inflammatory responses [35] and *S.* Typhimurium SL1344 mutants lacking *rhs* genes are completely attenuated in pig and cattle infection models [36], indicating a strong role for these systems in bacterial pathogenesis. In addition, removal of only the orphan toxin alone reduced *S.* Typhimurium proliferation in mice [33].

We were interested in finding out more about the role of Rhs toxins in *S.* Typhimurium biology. According to a genome-wide survey of the transcriptional response to infection-relevant conditions, multiple promoters were identified in the *rhs* locus (Fig 1A)[37], suggesting that while the full-length *rhs* gene was expressed rarely or not at all, strong expression of the main and orphan C-terminal toxins occurred at most conditions. We therefore set out to investigate if this expression resulted in functional toxin(s) being produced and if these toxins had a biological function in the cell. Here we show that main and orphan *rhs* genes encode functional type-II TA-modules that regulate *S.* Typhimurium proliferation inside macrophages.

## Results

### *rhs* toxins are expressed from internal promoters in the *rhs* ORF

According to a genome-wide survey of the transcriptional response to an array of 22 infection-relevant conditions encountered by *S.* Typhimurium 4/74, the main *rhs* gene (STM0291) encoding the Rhs shell and the associated T6SS genes are not transcribed during any condition (Fig 1A) [37]. However, the C-terminal toxin domains of both the main and orphan *rhs* genes, *rhs-CT*, as well as their cognate immunity genes, *rhsI*, were transcribed in several of the conditions assayed, including Early Stationary Phase (ESP), a condition which stimulates expression of the *Salmonella* pathogenicity island -1 (SPI-1) virulence genes and during intra-macrophage growth (MAC) (Fig 1A). To validate this expression, we performed quantitative real-time PCR using oligos that bind within the *rhs*$^{delivery}$, the main *rhs-CT* (*rhs-CT*$^{main}$) or the orphan *rhs-CT* (*rhs-CT*$^{orphan}$) (Fig 1A, Table A in S1 Text). Contrary to the transcriptomics data, the qPCR results suggest a small but significant expression of *rhs*$^{delivery}$. The *rhs-CT*$^{main}$ and *rhs-CT*$^{orphan}$ are, however, expressed 100-fold higher than *rhs*$^{delivery}$ in ESP and 5 to 9-fold higher respectively in MAC conditions (after 8h of growth) (Fig 1B and 1C).

Transcriptional profiling revealed four transcriptional start sites (TSS's) in the *rhs* locus region (here called P2-P5) (Fig 1A). Two of these TSS's are located in a conserved region found in both the main and orphan *rhs* genes (P2$^A$ and P2$^B$). These TSS's are located 89 nt upstream of the sequence encoding the conserved PxxxxDPXGL motif, which suggests that they could hypothetically generate an mRNA that would allow for the translation of a functional toxin and antitoxin. The other three TSS's identified were unique to either the main or orphan *rhs* genes. To test whether these TSS's (P2- P5) and an additional hypothetical TSS, upstream from the start codon of the *rhs*$^{delivery}$ gene (P1) allow for transcription, we constructed transcriptional fusions with sYFP2 to these promoters on a pSC101 vector backbone (S1 Fig). We defined the promoter region as -100 nt from the hypothetical TSS, except for P1 where we decided to include 200 nt in the transcriptional constructs. These constructs were transformed into *S.* Typhimurium 14028S and fluorescent activity was assayed in over-night cultures (grown for 16–18 h) by flow cytometry. In line with the published transcriptomic dataset we found the highest fluorescent activity for the P2 constructs in LB (Fig 2A and 2B). Also promoters P4 and P5 were active in LB, whereas we could detect no activity for P1 and P3 (Fig 2A and 2B). We chose to focus the rest of this study on the P2$^A$/P2$^B$ and their associated

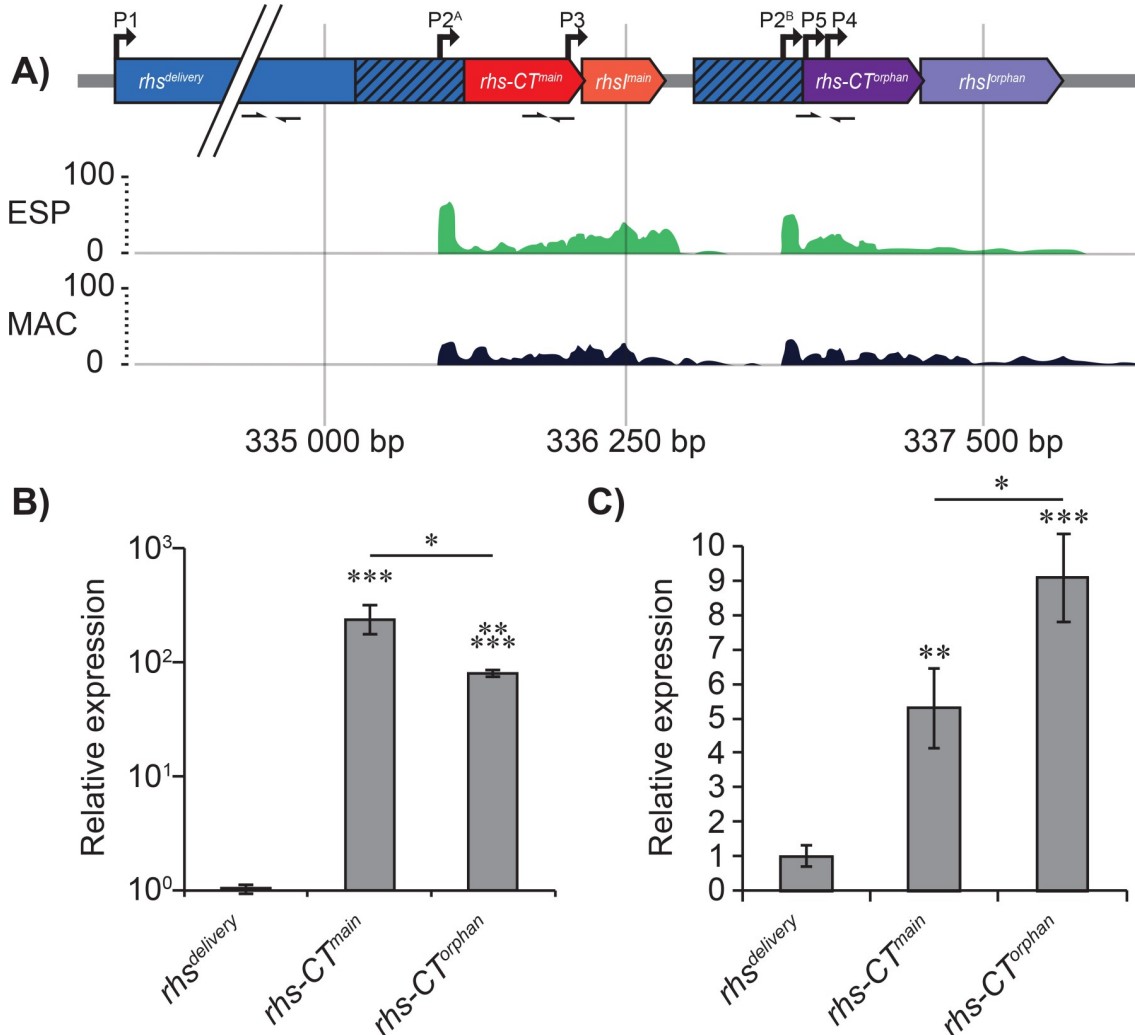

**Fig 1. Transcriptional activity of the *rhs* locus. A)** Illustration showing transcriptional activity from the SalCom database. Predicted transcription start-sites are shown as arrows above the schematic gene. **B-C)** Relative expression of *rhs*delivery, *rhs-CT*main and *rhs-CT*orphan determined by qPCR in **B)** LB broth or **C)** in RAW 264.7 macrophages. Error-bars are SEM. Statistical significance was determined using Mann-Whittney test where * = P<0.05, ** = P<0.01 and *** = P<0.001, **** = P<0.0001, ***** = P<0.00001.

main and orphan *rhs* transcripts. The identified P2$^A$/P2$^B$ TSS was confirmed by 5'RACE and bioinformatic analyses identified putative -10 and -35 regions upstream of the TSS (S2 Fig). In addition, putative ribosome binding sites (RBS's) preceding several putative translation start sites (CTG (ORF1) or ATG (ORF2)) were identified in both the main and orphan transcripts (S2A and S2B Fig). These data suggest that the Rhs toxins could be expressed internally, similarly to type II TA-modules.

## Internal transcription results in toxic proteins

Transcriptional activity from the *rhs-CT*main and *rhs-CT*orphan from P2$^A$ and P2$^B$ does not necessarily infer that a functional toxic protein is generated from these transcripts. For a functional toxin to be expressed, a translation start site is required. The downstream region of the confirmed TSS's contained two potential translation start sites (S2A and S2B Fig). To investigate if either of these translation start sites produced a toxic protein, we cloned the putative

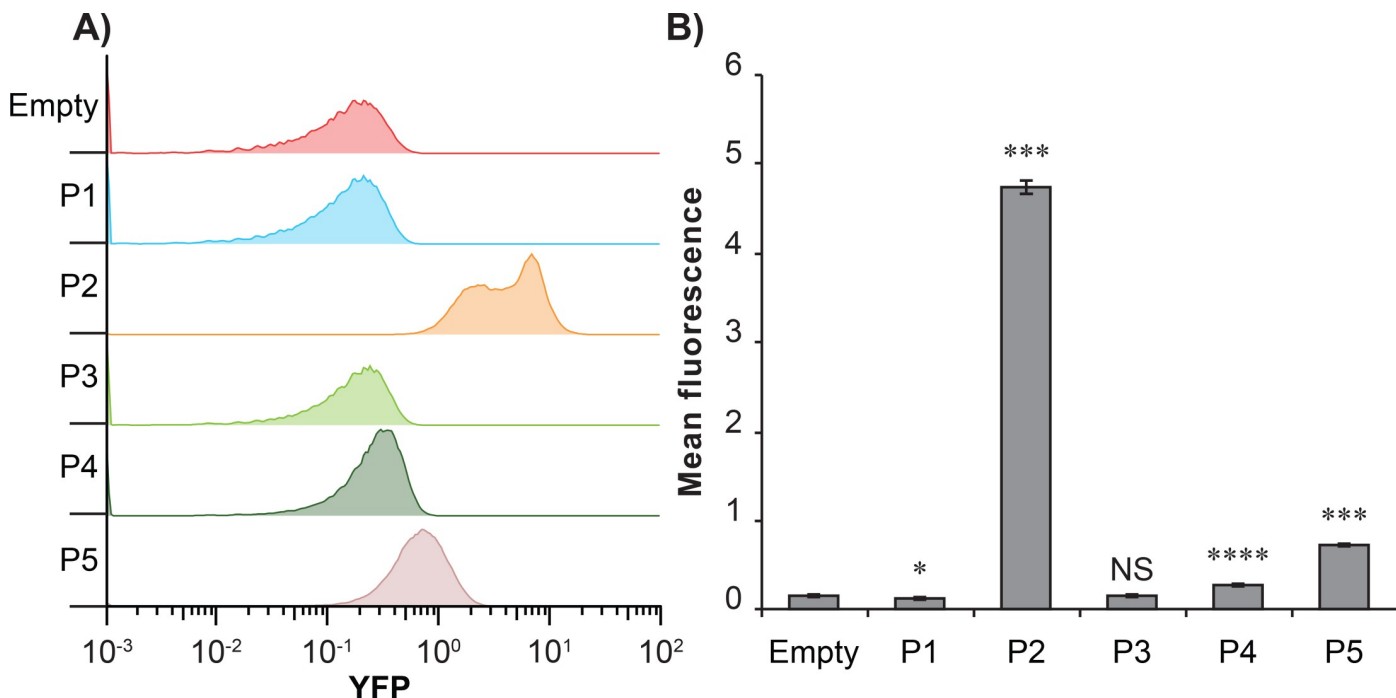

**Fig 2. Transcriptional activity of P1-P5 promoters using transcriptional fusions.** Transcriptional activity of P1-P5 promoters measured using fluorescence reporters with P1-P5 promoters to sYFP2. **A)** Representative flow graphs or **B)** mean YFP fluorescence of strains carrying pSK945-949 (P1-P5) or pEH167 (empty). Error-bars are SEM. Statistical significance was determined using two-tailed students T-test where * = P<0.05, ** = P<0.01, *** = P<0.001 and **** = P<0.0001.

ORFs (from the predicted start codon to the stop codon) under an arabinose inducible promoter on a p15A vector backbone. Toxicity of the internal transcripts was determined by transformation into NEB high-efficiency chemically competent cells in the presence or absence of the inducer (L-arabinose) and the ability of the cognate immunity protein to rescue this toxic effect. NEB cells transformed with plasmids expressing the longer ORF1 of Rhs-CT$^{main}$ or Rhs-CT$^{orphan}$ (from CTG) grew unaffectedly in glucose, with or without expression of their cognate immunity proteins, whereas 3- and 1.5- orders of magnitude less growth was observed in L-arabinose in the absence of cognate immunity proteins for the Rhs-CT$^{main}$ and Rhs-CT$^{orphan}$ toxin, respectively (Fig 3). In contrast, the growth of NEB cells transformed with plasmids expressing the shorter ORF2 of Rhs-CT$^{main}$ or Rhs-CT$^{orphan}$ was unaffected in the absence of cognate immunity when grown in either arabinose or glucose (S3 Fig). We concluded that translation of a functional toxin, originating from the CTG start codon (ORF1) was possible from the P2$^{A}$ transcript and that this toxin could be inactivated by the presence of its cognate immunity protein in the cell.

Cells expressing adequate levels of cognate immunity protein are protected from the toxicity of C-terminally encoded toxins of CdiA and Rhs molecules [20, 29, 34]. For CdiA toxins, the immunity proteins neutralize the toxins by binding the toxin directly, similarly to known type-II-TA-systems [38]. To investigate if also RhsI mediates protection through direct interaction with the toxin we used co-immunoprecipitation. HA-tagged RhsI$^{orphan}$ was detected in cell lysates of cells expressing *rhsI* $^{orphan}$-HA as well as in immunoprecipitated samples using anti-HA magnetic beads (Fig 4A). Similarly, Rhs-CT$^{orphan}$ was present in all whole cell lysates of strains expressing *rhs-CT$^{orphan}$*, but could only be detected in the immunoprecipitated fractions of cells expressing *rhsI* $^{orphan}$-HA (Fig 4B). Co-immunoprecipitation of Rhs-CT$^{orphan}$ and

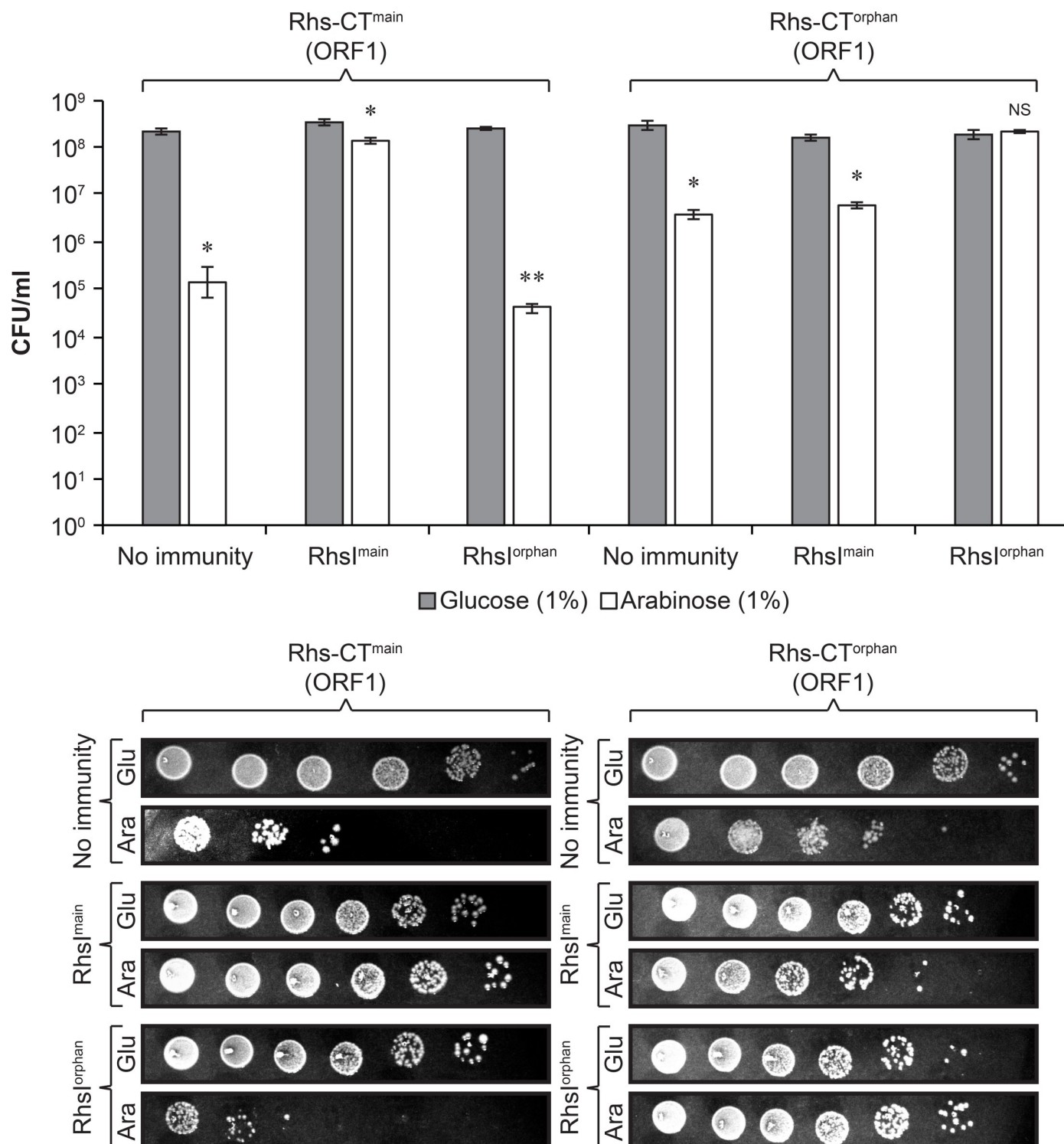

**Fig 3. Toxicity of Rhs-CT^main and Rhs-CT^orphan.** Colony forming unit (CFU) values of NEB 5-α cells with or without immunity plasmid (No immunity/*rhsI*^main pDAL923 /*rhsI*^orphan pDAL938) 21h after transformation with plasmids encoding arabinose inducible *rhsCT*^main pSK1913 /*rhsCT*^orphan pSK1914 (ORF1) grown on either 1% glucose or 1% arabinose. n = 3, Error-bars are SEM. Statistical significance was determined using two-tailed students t-test where * = P<0.05 and ** = P<0.01.

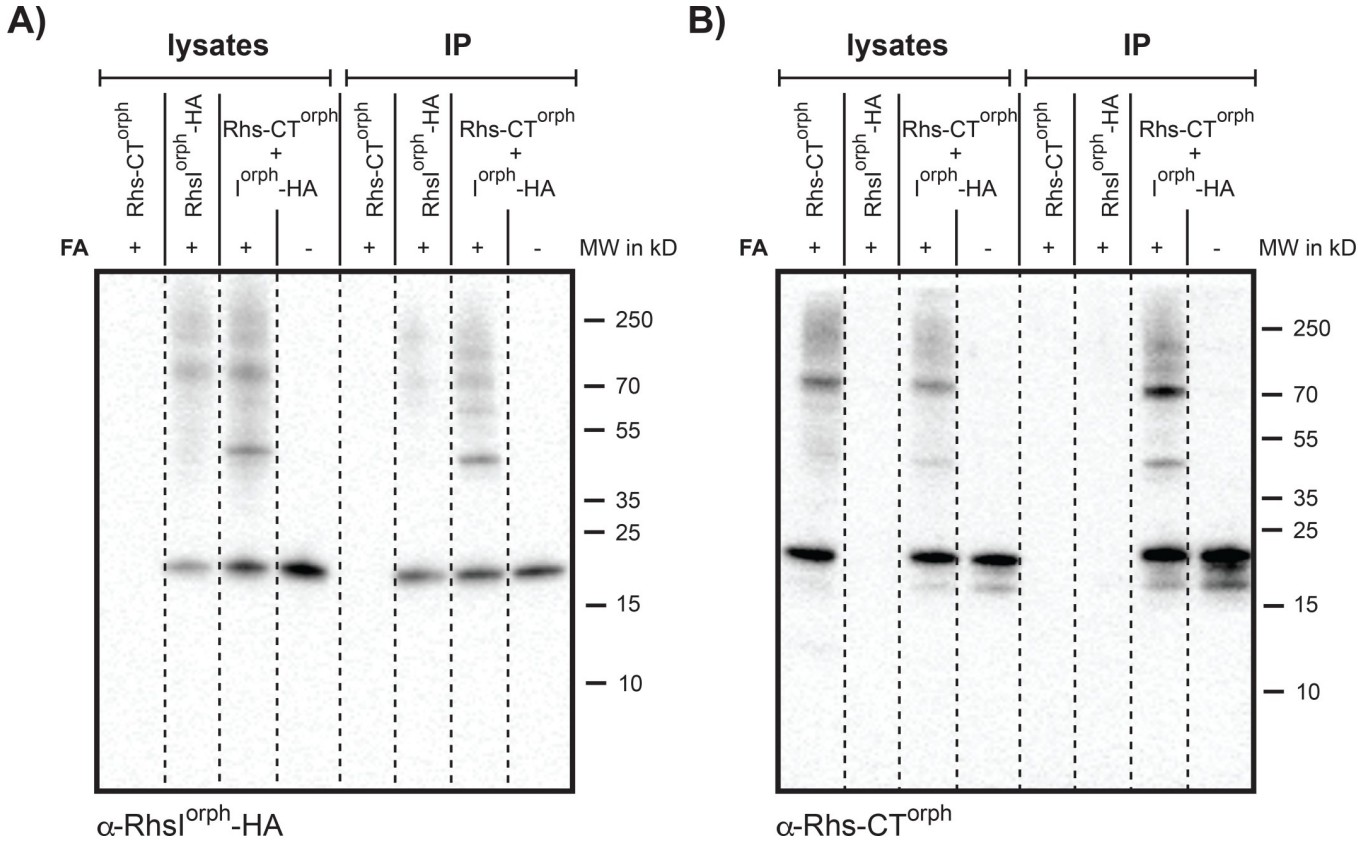

**Fig 4. RhsCT$^{orph}$ and RhsI$^{orph}$-HA interact upon co-expression.** Immunoblot analysis of cleared cell lysates and immunoprecipitated samples (IP) using anti-HA magnetic beads with (+) or without (-) formaldehyde treatment (FA). Cells contain either RhsCT$^{orphan}$ expressed from an arabinose inducible plasmid (pSK502) or RhsCT$^{orphan}$ and RhsI$^{Orphan}$-HA expressed from an arabinose inducible plasmid (pSK4251). **A)** RhsI$^{Orphan}$-HA was detected with anti-HA antibodies. **B)** RhsCT$^{orphan}$ was detected using purified anti-RhsCT$^{Orphan}$ antiserum.

RhsI$^{orphan}$-HA was observed both in cell lysates treated with formaldehyde prior to lysis and in untreated cell lysates. This suggests that the Rhs orphan immunity neutralizes Rhs orphan toxin through direct interaction as observed for other CDI and type II-TA-systems.

The immunoprecipitations were done with a plasmid where the entire *rhsCT-I*$^{orphan}$ locus was cloned with its native translation starts. To be able to detect RhsCT$^{orphan}$ by western blot, we had to over-express an inactive mutant of this toxin from a previously described vector [34]. This is not surprising as native levels of expression (from the CTG start codon) are expected to be much lower than from the artificial ATG start included in the over-expression construct. To see if the internal transcription also was able to produce protein from the native P2$^A$ and P2$^B$ transcripts and the CTG translation start site, we constructed translational reporter fusions where sYFP2 was inserted 3 aa downstream of the CTG start codon in ORF1, shown to give a toxic product (Fig 3). Protein levels were investigated by single molecule fluorescence microscopy. Using this construct, a small, yet significant level of YFP was detected from Rhs-CT$^{orph}$, but not Rhs-CT$^{main}$. (S4A Fig), suggesting that both internal transcripts could result in protein although at levels below the detection limit for the native Rhs-CT$^{main}$. To further investigate if we could observe a signal from Rhs-CT$^{main}$ we repeated the experiment at a higher laser voltage and longer exposure time. At these settings, a significant level of expression could be observed from the native Rhs-CT$^{main}$ (S4B Fig).

## Rhs toxins repress *S.* Typhimurium proliferation within host macrophages

Previous studies have shown that TA-systems are important for *S.* Typhimurium growth regulation within macrophages [1]. Our results suggest that the Rhs-CT$^{main}$ and Rhs-CT$^{orphan}$ toxins are expressed intracellularly similar to type-II TA-systems, but whether this has an effect on bacterial physiology is unclear. To test if Rhs toxins were important for *S.* Typhimurium proliferation within macrophages, we infected RAW264.7 macrophage cells with wild type or mutant *S.* Typhimurium bacteria lacking *rhs-CT*$^{main}$ (Δ*rhs-CT*$^{main}$), *rhs-CT*$^{orphan}$ (Δ*rhs-CT*$^{orphan}$), *rhs-CT*$^{main}$ and *rhs-CT*$^{orphan}$ (Δ*rhs-CT*$^{main+orphan}$) or the entire *rhs*-locus (Δ*rhs*$^{complete}$) (Fig 5A). To reduce fluctuation between biological replicates due to variation in the number of RAW 264.7 cells in individual wells, we labelled wild type *S.* Typhimurium with one antibiotic resistance marker (CAM) and wild type/mutant with another antibiotic resistance marker (KAN) and infected each well with a mixture of wild type/wild type or wild type/mutant *S.* Typhimurium. The survival of wild type and wild type/mutant bacteria was monitored by plating cell lysates on LB agar plates containing either KAN or CAM antibiotics. This allowed us to calculate an accurate intramacrophage survival of our mutants relative to wild type for each biological replicate. Relative intramacrophage survival was calculated as the ratio of colony forming units (CFU) on KAN/CAM at 15h divided by the same ratio at 0h. To avoid competition and potential Rhs toxin delivery between wild type and mutants during the infection experiment, we used a low multiplicity of infection (MOI = 10) to ensure that only one infecting bacterium would be found per macrophage. Interestingly, we find that the Δ*rhs-CT*$^{main+orphan}$ and Δ*rhs*$^{complete}$ mutants proliferate approximately 2-times better within the macrophages than wild type *S.* Typhimurium (Fig 5B), suggesting that the Rhs toxins slow down or arrest growth of *S.* Typhimurium within macrophages. To control that no other mutations (that could have accumulated during strain construction) affected the growth of *S.* Typhimurium in our assay, we whole genome sequenced the Δ*rhs*$^{complete}$ mutant. (The Δ*rhs*$^{complete}$ mutant was generated by P22 transduction into the Δ*rhs-CT*$^{main+orphan}$ mutant, which in turn was generated by P22 transduction into the *rhs-CT*$^{orphan}$ mutant. Thus, the *rhs-CT*$^{orphan}$ and Δ*rhs-CT*$^{main+orphan}$ mutant should not contain additional mutations). Whole genome sequencing revealed 3 additional SNPs in wild type *S.* Typhimurium LT2 (listed in Table B in S1 Text). These mutations reflect the fact that our genetic engineering was performed in *S. typhiumurium* LT2 after which the constructs were moved to 14028S by P22 transduction. To ensure that these SNPs were not contributing to the observed phenotype of our mutants, we constructed a complemented strain where the entire Rhs-locus was reconstituted in its native genetic location in the Δ*rhs*$^{complete}$ mutant (carrying all three SNPs). No increase in survival could be observed in this mutant (Fig 5B).

Previous studies indicate that only a subpopulation of *S.* Typhimurium cells will start to replicate during infection and that a large fraction remains non-growing [39]. Thus, the observed outgrowth could be the result of two different scenarios where either; i) a larger fraction of mutant bacteria resume growth after entry into the macrophages, ii) the bacteria that resume growth proliferate faster within the macrophage or a combination of the above. To test which of these scenarios was the case we used a previously described fluorescent dilution assay, where the bacteria were transformed with a plasmid constitutively expressing GFP and where RFP (dsRed) expression is under the control of an arabinose inducible promoter. By expressing RFP prior to infection (by growing the bacteria in media supplemented with arabinose), the fraction of growing *S.* Typhimurium cells within the macrophages can be detected as the RFP signal will be diluted by 50% with each cell division [39]. The fraction of growing cells was calculated as 1-fraction non-growing cells (S5 Fig) and adjusted to account for the number of generations the growing bacteria had undergone during the experiment (for

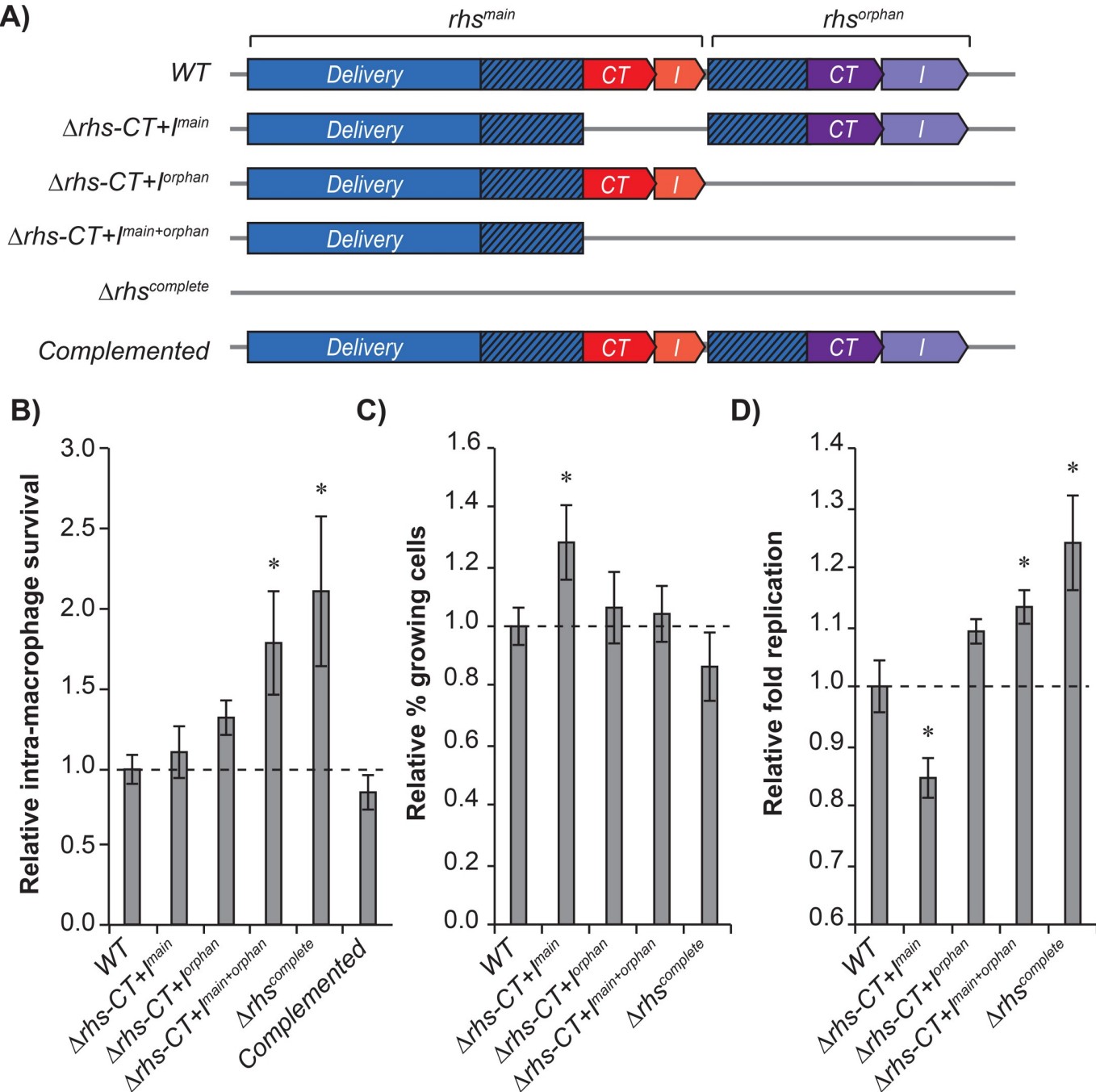

**Fig 5. Internal expression of *rhs-CT*main and *rhs-CT*orphan reduce *S*. Typhimurium growth in RAW 264.7 macrophages. A)** Illustration showing the genetic mutants used in **B-D. B)** Relative intra-macrophage survival of wild type, Δ*rhs-CT*main, Δ*rhs-CT*orphan, Δ*rhs-CT*main+orphan, Δ*rhs* complete mutants after 16 h. Relative survival was calculated as ratio of mutant/wild type at 16h as compared to the same ratio at 0 h. **C)** Average wild type, Δ*rhs-CT*main, Δ*rhs-CT*orphan, Δ*rhs-CT*main+orphan, Δ*rhs*complete mutants that starts to replicate after 16 h in RAW 264.7 macrophages relative to wild type. **D)** Relative growth rate of wild type and *rhs* mutants during 16 h infection. Error-bars are SEM. Statistical significance was determined using Mann-Whittney test where * = P<0.05.

calculations see methods section). Surprisingly, no significant differences could be observed in the population of bacteria that resumed growth after entry into the macrophages in wild type or mutant *S*. Typhimurium (Fig 5C). In contrast, 27% more bacteria started to replicate in

macrophages in *S.* Typhimurium lacking *rhs-CT*^main (Δ*rhs-CT*^main), suggesting that this mutation affected the proportion of cells resuming growth. The increased proliferation observed for the other mutants could, however, not be accounted for by an increased fraction of cells resuming growth. To investigate if instead the mutant bacteria grew faster within the macrophage, we calculated the fold replication and number of generations the bacteria underwent during the time-frame of the experiment. The fold replication bacteria undergo during 16h of growth within macrophages can be calculated as the change in median RFP intensity between the initial population (0h) and the growing population after 16h (S5 Fig). The relative fold replication was then calculated as the fold replication of the mutant over wild type. *S.* Typhimurium cells lacking the entire *rhs* operon grew ~30% faster than wild type during the experiment, allowing the mutant bacteria to divide one extra time during the experiment (~7 generations compared to ~6 generation) (Fig 5D). One extra division could account for the 2-fold increase in CFU observed in Fig 5B. Similarly, for the single *rhs-CT*^orphan and the double *CT* deletion mutants grew 20% faster than wild type (Fig 5D). For the *rhs-CT*^main deletion mutant no increase in growth rate could be observed, suggesting that the increase in fraction of cells resuming growth must account for the increase in CFU. Taken together, our results suggest that the internal expression of the Rhs toxins contribute mainly to the growth rate of *S.* Typhimurium after entry into macrophages and not the fraction of cells resuming growth.

**RhsI antitoxins are degraded by proteases.**   A remaining question is how Rhs toxins arrest growth. A plausible hypothesis is that they function similarly to toxins belonging to type II TA-systems and CdiA toxins, where the antitoxin is degraded by a stress-induced protease; Lon [40, 41]. Lon is up-regulated in *S.* Typhimurium after entry into macrophages [42], suggesting a faster turnover of antitoxin degradation in this condition. This could result in that cells with the Rhs TA-system arrest growth because they lack sufficient levels of antitoxin, whereas mutant cells lacking the TA-system can grow less hindered. To test if RhsI was susceptible to degradation by the Lon protease, we cloned C-terminally HA-tagged versions of the RhsI proteins under an arabinose inducible promoter expressed from the pBAD33 vector. Protein stability was measured in wild type and Δ*lon* bacteria using western blot with anti-HA antibodies after the addition of either DL-Serine hydroxamate (SHX), a seryl-tRNA synthetase inhibitor, or the macrolide antibiotic Erythromycin, an inhibitor of protein synthesis, to cultures expressing RhsI^main or RhsI^orphan. The RhsI^orphan protein was stable over the course of 1h in Δ*lon* bacteria compared to wild type where degradation occurred already after 15 min after SHX addition (Fig 6A, lower panel). In contrast, the RhsI^main protein was degraded 30 min after SHX addition in both wild type and Δ*lon* bacteria, suggesting that another protease is responsible for the degradation of this protein (Fig 6A, upper panel). Both proteins were completely stable in both wild type and Δ*lon* background upon Erythromycin treatment (Fig 6B). These results suggest that Rhs immunities do not have a fast turn-over under favorable conditions but are unstable and subject to proteolytic degradation when the cell is stressed. From these results it appears that Rhs toxins could arrest growth upon entry to macrophages through stress-mediated activation of cellular proteases, resulting in RhsI antitoxin degradation.

## Discussion

RHS elements have previously been described as weapons used in antagonistic bacterial interactions. Here we show that Rhs toxins are not only being delivered between cells but are also expressed and functional within a bacterial cell. We also show that Rhs toxins affect proliferation of *Salmonella* cells during macrophage infection by changing the growth rate of *S.* typhimurium within macrophages. Thus, Rhs toxins comprise functional type-II TA loci that repress *S.* Typhimurium growth *in vivo*. Previous analysis of more than 150 *Salmonella*

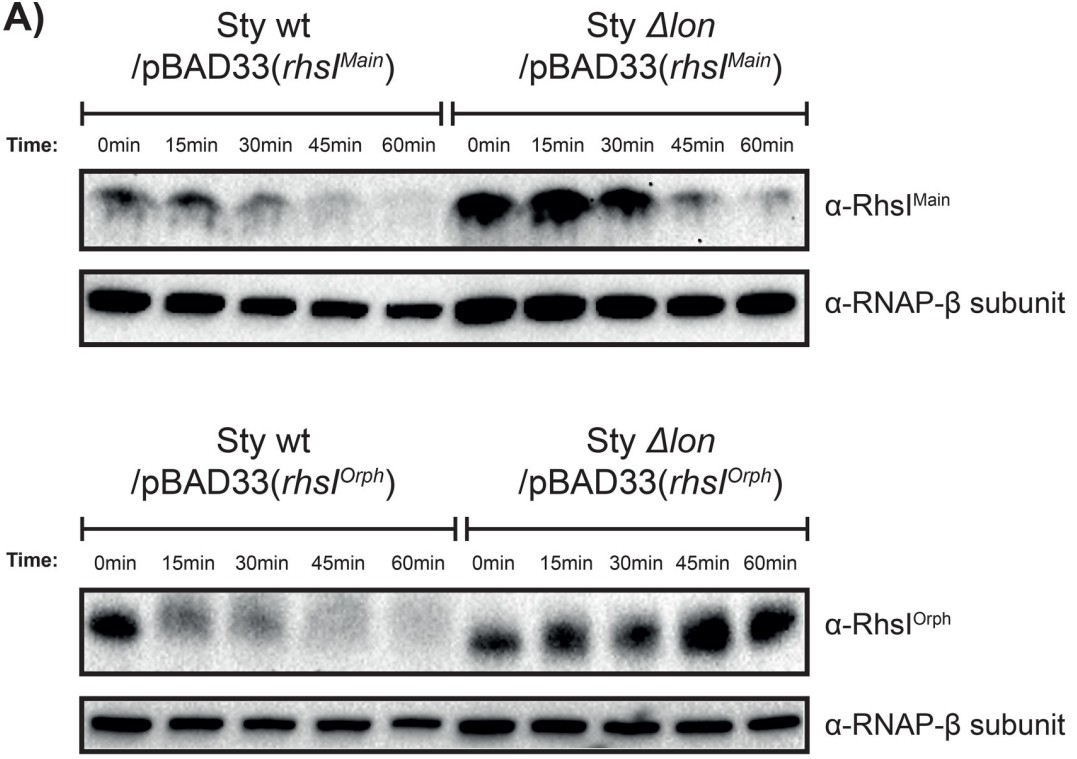

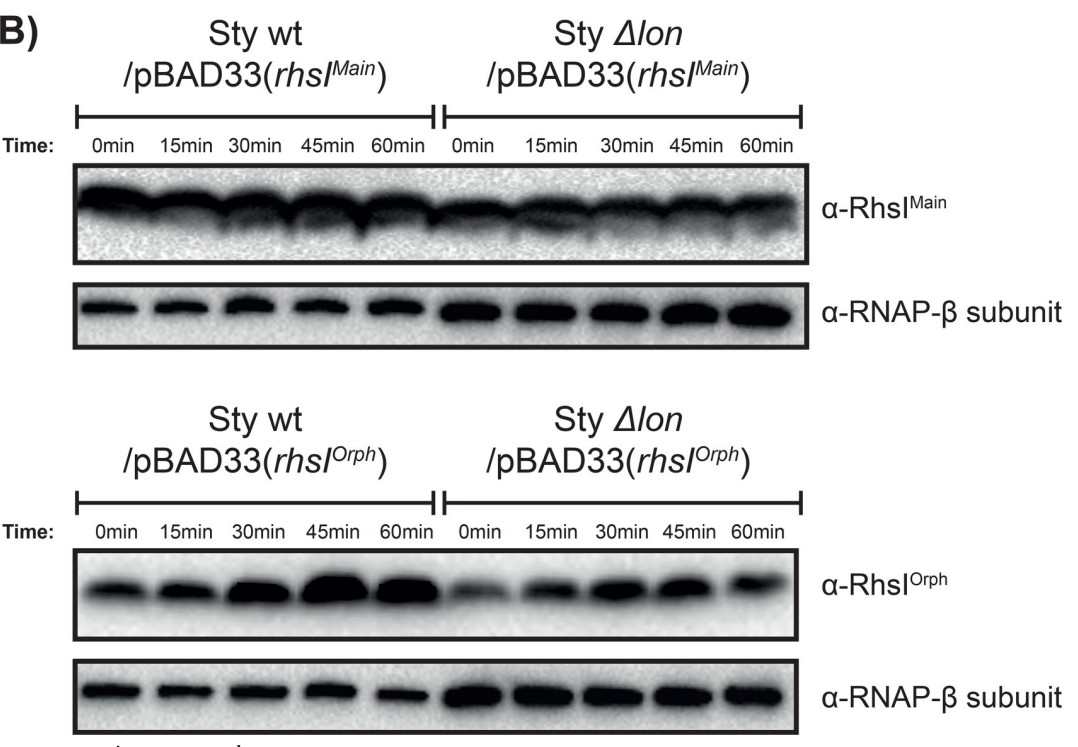

**Fig 6. RhsI^main and RhsI^orphan are degraded by proteases (including Lon) upon stringent-response activation.**
Immunoblot analysis of RhsI-HA^main or RhsI-HA^orphan protein (**11.6 and 18.5 kDa respectively**) stability in the presence (Sty

wt) and absence (Sty Δ*lon*) of Lon protease before and after **A)** SHX or **B)** Erythromycin treatment for 1 h. Levels of the RNA polymerase β-subunit are shown as control.

genomes showed that all *S*almonella serovars contain at least one Rhs toxin, with some strains having up to 11 different orphan toxins [34]. Bioinfomatic analyses of all genomes on NCBI showed that the P2 promoter is found upstream of at least 50 different Rhs toxins in *Salmonella* spp. but also in *E. coli*, *Shigella*, *Pseudomonas* and *Erwinia* spp, suggesting that also these toxins could be expressed internally as seen in *S*. Typhimurium (S6 Fig). This raises an interesting question; does the presence of other Rhs toxins affect *Salmonella* growth *in vivo* and, if so, what are the consequences of having up to 12 toxins for *Salmonella*'s lifestyle and ability to replicate in macrophages? On the other hand, it is not obvious why *S*. Typhimurium has two Rhs toxins, when one alone (the orphan) has a strong effect on intramacrophage replication. A possible explanation could be that the two toxins are used for different things. For example, the main Rhs toxin can be delivered between bacteria, whereas mobilization of the orphan toxin requires a pre-existing duplication of the locus followed by a recombination event [34]. Our results here suggest that the orphan toxin mainly functions in controlling intramacrophage replication, whereas very little contribution can be seen from the main Rhs toxin. It is also possible that the contribution of *rhsCT* $^{main}$ cannot be fully evaluated in the single mutant as removal of *rhsCT* $^{main}$ allows for a recombination event that will fuse *rhsCT* $^{orphan}$ to the *rhs*-delivery mechanism [34]. Such recombination events between hundreds of base pairs of exact homology are very frequent [43], but are prevented in wild type *Salmonella* by the internal expression of RhsCT/I main (which upon loss would cause a growth defect to that cell). Changing the chromosomal organization of the locus could result in a number of changes including altered expression levels of the RhsCT$^{orphan}$ as well as potential delivery of the toxin between cells, which could affect proliferation in our infection assay.

So, what could be the benefit of maintaining these Rhs toxins in *Salmonella* genomes? One possible explanation is that there is no benefit, but that Rhs toxins represent selfish elements and are expressed internally simply to maintain selection of themselves. This fits nicely with the observation that these internal promoters are expressed under all conditions tested and could also explain why Rhs proteins are some of the most positively selected genes known [44]. Selfish genetic elements can either consist of sequences that frequently transfer between strains but can also consist of elements that provide a benefit to the cell, or a combination of both [45]. These selfish genetic elements may have been domesticated or co-opted by the bacteria to provide positive effects for the cell, like defense against phage, or ability to compete with other bacteria. The Rhs toxins in *S*. Typhimurium can be mobilized and used to outcompete bacteria lacking these systems [34]. But, the toxins are found on the H-NS silenced SPI-6 [46], previously shown to be important for colonization of mice [33]. Thus, it is possible that the internal expression of Rhs toxins provides a selection to maintain the full-length Rhs toxin and/or SPI-6 in conditions when they are not expressed. Another possibility is that growth arrest after entry into macrophages could be important for *S*. Typhimurium *in vivo*. Replication in macrophages occurs mainly during systemic spread of the disease within the lymph nodes, liver and spleen. Interestingly, a previous study showed that Rhs-CT$^{orphan}$, but not RhsI$^{main}$ is important for systemic dissemination in mice [33], suggesting that internal expression could be important for virulence. *Salmonella* gastroenteritis infections are often self-limiting and do not proceed beyond the lamnia propria. However, during typhoid fever, the *Salmonella* infected phagocytes gain access to the lymphnodes, liver and spleen [47]. It is possible that regulation of growth upon entry to phagocytes is an important factor in deciding whether the infection stays local or becomes systemic.

The full-length Rhs toxins or the adjacent T6SS genes are not expressed during any of the conditions assayed in Kröger *et al.* [37]. Previous findings suggest that SPI-6 is strongly silenced by H-NS and that only in conditions where H-NS expression is decreased or absent can T6SS activity be observed [32, 46]. Other contact-dependent competition systems have been suggested to contribute to kin selection by exclusion of others (by inhibiting the growth of non-self bacteria) [48]. Rhs toxins have also been shown to be important during intestinal colonization [32, 49]. As the intra-macrophage population most likely is clonal, there is probably no competition with non-kin cells in this niche. Thus, expression of a system that is only used as a competition system, in these conditions, could be considered a waste of resources. We cannot, however, exclude the possibility that the ability to deliver toxins between isogenic strains adds to the growth regulation observed in our experiments. We have previously shown that self-delivery of CdiA toxins increases the fraction of non-growing cells in the population [41], so it is possible that Rhs toxins expressed at a low level (undetectable by RNA seq) are delivered between isogenic cells within the phagocytic vesicle. This could potentially explain the increased effect observed in the complete mutant as compared to the one lacking both toxins.

A remaining question is how Rhs toxins arrest growth. Our results suggest that both RhsI proteins are susceptible protease mediated degradation under conditions when translation is arrested by induction of the stringent response, but not when translation is arrested by antibiotics. For one of the immunity proteins, RhsI$^{orphan}$, this degradation was completely abolished in cells lacking Lon protease. Lon protease becomes activated by poly-phosphate under conditions when stringent response is active [50, 51], suggesting that activated Lon is required for degradation of RhsI$^{orphan}$. As Lon is up-regulated in *S.* Typhimurium after entry into macrophages [42], it is likely that antitoxin turnover is increased in this condition. This would result in that cells with the Rhs TA-system arrest growth because they lack sufficient levels of antitoxin, whereas mutant cells lacking the Rhs TA-system can grow unhindered. Previous findings show similar results where lack of certain toxin-antitoxin pairs decrease the non-growing *Salmonella* population within macrophages [1]. It is likely that each individual toxin adds a little to the effect and it would be interesting to see how a combined mutant lacking both type II-TA-systems and Rhs toxins would behave during macrophage infection. Interestingly, we observe a stronger phenotype *in vivo* for the Rhs$^{orphan}$ toxin even though it is less toxic than Rhs$^{main}$ when expressed from an arabinose inducible promoter (as seen in Fig 3). A possible explanation is that the two immunity proteins are not degraded to the same extent or at the same time in the macrophages. RhsI$^{orphan}$ is degraded by the Lon protease, whereas another protease, possibly not activated to the same extent or at the same time within the macrophage, degrades RhsI$^{main}$ (as seen in Fig 6). Protease mutants are non-viable in macrophages [52], making it difficult to test how protease mediated degradation of the antitoxin affects *Salmonella* growth within macrophages.

An interesting and logical follow up question is of course how the growth arrested cells are able to resume growth? As with many type II toxins, CDI toxins are rarely lethal, depending on their toxic activity. It is easy to envision that cells undergone toxicity that results in degraded RNA molecules or dissipation of the proton motive force, easily can resume growth once the toxicity has been removed through degradation or neutralization of the toxin by proteases or newly synthesized antitoxin respectively. The toxic activity of the Rhs toxins in *S.* Typhimurium is not known. Structural or functional predictions suggest no membrane association for either the toxins or immunities, implying that the Rhs toxins do not act on the membrane. PFAM database search reveals a known toxin domain (NTox) in the Rhs$^{main}$ toxin, suggesting that this protein is an RNAse [53]. For Rhs$^{orphan}$ toxin, no clues to its activity could be revealed from its sequence.

Orphan toxins that lack the secretion component of the toxin are not only found for Rhs toxins, but are also often present in *cdiBAI* loci [54]. A fundamental question is how these orphan modules are maintained in bacterial populations. One possible selection pressure could be that these toxins provide a silent arsenal of novel toxins that can be mobilized by bacteria and used in future competitions against neighboring bacteria. For example, this would facilitate a clonal selection in a population of genetically identical bacteria [34]. Our findings that orphan Rhs toxins represent functional TA-modules expressed inside the bacteria provides an additional explanation to how these orphan toxins can be maintained when not used for competition. The remaining question is whether also CdiA toxins, colicins, microcins and other T6SS toxic effector proteins also contain internal transcription and translation start sites allowing them to be expressed internally, and if that expression would have a function on said bacteria's physiology.

## Methods

### Bacterial strains and growth conditions

Bacterial strains used in the study are derivatives of either *Salmonella enterica* subsp. *Enterica* serovar Typhimurium strain LT2 or strain 14028S (ATCC 14028) and are listed in Table C in S1 Text. Liquid media was Luria-Broth (LB) (Miller, 1972). Over-night cultures were grown for 16–18 hr at 37˚C and 200 rpm. For solid media LB was supplemented with 10 g/L agar. Antibiotics were used at the following concentrations; 50 mg/l Kanamycin (KAN), 15 mg/L Tetracycline (TET), 12.5 mg/L Chloramphenicol (CAM), mg/L Carbenicillin (CARB) and 100 mg/L Ampicillin (AMP). Genetic mutations in the chromosome were made using λ red recombineering [55]. Markers were moved between *Salmonella* strains using P22 transductions. Transductants were purified on EBU plates [56] and clones were confirmed by PCR and sequencing. Antibiotic resistance markers were excised using Flp recombinase expressed from the pcp20 vector as described previously [55]. More detailed information about strain constructions can be found in the S1 Text.

### Expression of transcriptional fluorescent protein reporters measured through flow cytometry

Bacteria from over-night cultures were washed 2x with 1 ml sterile-filtered PBS and diluted to $1x10^6$ CFU/ml in PBS. Flow cytometry was performed using a MACSQuant VYB (Miltenyi Biotec), measuring 100,000 events. YFP was excited with a blue laser (488 nm; bandpass filter 525/50 channel B1), dsRed was excited with a yellow laser (561 nm; bandpass filter 615/20 nm channel Y2). The MACSQuantifyTM Software (Miltenyi Biotec) was used to aquire the data and the FlowJo Software (FlowJo, LLC) for analysis.

### 5'RACE

5' RACE (rapid amplification of cDNA ends) was carried out with or without treatment by 5′→3′ exoribonuclease XRN-1 (NEB), using DNase I-digested total RNA isolated from early stationary phase (ESP). RNA was ligated with an RNA adaptor (5' CGACUGGAGCACGAG GACACUGACAUGGACUGAAGGAGUAGAAA) and the product was reverse transcribed to cDNA using gene-specific oligo SK332 and Maxima first strand cDNA Synthesis Kit for RT-qPCR (K1641, Thermo Scientific). The DNA fragment was amplified in two consecutive rounds of PCR with DreamTaq DNA polymerase (ThermoScientific), first step with adapter-specific oligo FS17 and in the second, nested step with PCR oligo FS354, both in combination with gene specific reverse oligo SK332. The XRN-1-enriched DNA fragments were gel purified

and cloned into TOPO vector pCR-BLUNT-II (Thermofisher), and 10 clones were sequenced to validate the transcriptional start site.

## Western blot

500 μl of an $OD_{600}$ = 0.7 culture of exponentially growing *S.* Typhimurium cells, induced with 0.2% arabinose at $OD_{600}$ = 0.3, were harvested by centrifugation at 4˚C for 5 min at 21000 x g. Cells were re-suspended in 100 μl of sample buffer (50mM Tris pH 6.8, 1% SDS (w/v), 1% Triton X-100, 10% Glycerol, 0.2% bromophenol blue) and boiled for 5min at 95˚C. 150mM DTT was then added to each sample followed by the pelleting of membrane debris and undigested genomic DNA at 21000 x g for 5 min before the supernatant were separated on a Mini-PROTEAN TGX gel (Biorad, USA). PageRuler Prestained Protein Ladder (Thermo Scientific, USA) was used as size marker. Proteins were then transferred to a Trans-Blot Turbo Mini 0.2 μm PVDF membrane (BioRad, USA) using the Trans-Blot Turbo system (Biorad, USA). RhsI$^{main}$-HA and RhsI$^{oprhan}$-HA proteins were detected using anti-HA antibody (11867423001, Sigma-Aldrich, Germany). Equal loading was confirmed by probing for RNA polymerase β-subunit using an anti-RNAP β-subunit antibody (ab191598, Abcam, United Kingdom). Secondary antibody towards anti-HA antibodies was anti-rat IgG coupled to HRP (GENA935, Sigma-Aldrich, Germany) and secondary antibody towards anti-RNAP antibodies was anti-rabbit IgG coupled to HRP (A1949, Sigma-Aldrich, Germany). Bands were visualized with Clarity Western ECL Substrate (Biorad, USA) and a ChemiDoc MP system (Biorad, USA).

## Co-Immunopurification (Co-IP)

Exponentially growing cultures of SK502 (pCH450::*rhsCT*$^{orphan\ (H208A)}$), SK4251(pBAD24:: *rhsCT-I*$^{orphan}$-HA), SK4292 (pCH450::*rhsCT*$^{orphan\ (H208A)}$ and pBAD24::*rhsCT-I*$^{orphan}$-HA) ($OD_{600}$ ~0.9–1.2) were induced with 0.5% L-arabinose (f.c.) at $OD_{600}$ = 0.3 for ~1h. 90 ml of cells were harvested by centrifugation and washed once with 1 x PBS. Cells were resuspended in 5 ml 1 x PBS supplemented with 2% formaldehyde (f.c.) and incubated for 10 min at RT. Residual formaldehyde was quenched by addition of 5 ml 1M Tris-Cl pH 8.0 and two subsequent washes with 0.5 M Tris-Cl pH 8.0. Cells were resuspended in lysis buffer (20 mM Tris-Cl pH 7.5, 0.1% Tween-20, 0.25% Triton-X-100, 1x Halt-TM EDTA-free protease inhibitor mix) and broken by 6 cycles of sonication for 10 sec at 50% amplitude on ice. Debris and remaining cells were removed by centrifugation at 21.000 x g for 15 min at 4˚C. Cleared cell lysates were incubated with Pierce$^{TM}$ anti-HA magnetic beads (Thermo Scientific) according to the manufacturer's instructions. Bound proteins were eluted with 50 μl 0.2 M glycine pH 2. Cleared cell lysates and eluates from the magnetic beads were dissolved in NuPAGE LDS sample buffer (Thermo Fisher Scientific) supplemented with 0.1 M dithiotreitol (f.c.) and incubated at 60˚C for 10 min prior to loading. Of each cleared cell lysate, a volume corresponding to 1ml of bacteria with $OD_{600}$ = 0.14 (for detection of RhsI$_{orph}$-HA) or $OD_{600}$ = 0.014 (for detection of RhsCT$^{orph}$) was separated on a 10% NuPAGE Bis-Tris gels (Novagen) using MES running buffer supplemented with antioxidant. Proteins were transferred to a PVDF-membrane using pre-assembled iBlot$^{TM}$ transfer stacks (Thermo Fisher Scientific). RhsCT$^{orph}$ was detected using purified anti-RhsCT$^{orph}$ antiserum [34] and anti-rabbit IgG coupled to HRP (Sigma). RhsI$^{orph}$-HA was detected using anti-HA-antibody as described above.

## RAW264.7 culturing conditions

The murine macrophage-like cell line RAW264.7 (ATCC) was cultured in 1x Dulbecco's Modified Eagle Media (DMEM, GIBCO) supplemented with 1% GlutaMAX (GIBCO), 1% non-essential amino acids (Sigma), 10% heat inactivated Fetal Bovine Serum (HI-FBS) (GIBCO) at

37°C with 10% $CO_2$ in T-25 bottles (Sarstedt). Cells were passaged when confluent or every 48 h.

### RAW264.7 infection assay

$1x10^5$/ml RAW264.7 cells were seeded in 24-well plates (Sarstedt) in pH buffered DMEM (high glucose, HEPES, no phenol red, GIBCO) supplemented with, 1% non-essential amino acids (Sigma) and 10% HI-FBS (GIBCO) and grown at 37°C with 10% $CO_2$ for 48 h before the infection. Over-night cultures of *S*. Typhimurium 14028S were centrifuged at 6000 RPM and washed with sterile 1x PBS before being diluted to $OD_{600}$ = 0.4. Bacteria were then opsonized in Balb/cAnCrl mouse serum for 30 min. The old growth medium was removed and fresh pH buffered DMEM without FBS was added before infection at MOI 10. The infection was synchronized by centrifuging the cells for 5 min at 1200 rpm followed by 45 min incubation at 37°C with 10% $CO_2$. The wells were washed with 1x PBS before adding fresh pH buffered DMEM + gentamicin (100 mg/l). Cells were incubated 45 min at 37°C to kill off extracellular bacteria followed by washing with 1x PBS before being lysed with 0.1% Triton-X solution (Sigma). Lysates were immediately plated for CFU counts of the 0 h time point. Fresh pH buffered DMEM + gentamicin (20 mg/l) was added to the remaining wells for later time points and incubated at 37°C with 10% $CO_2$. Relative intra-macrophage survival was calculated as the ratio of WT and mutant CFU/ml at 16 h divided by the ratio of WT and mutant CFU/ml at 0 h.

### Fluorescence dilution assay

*S*. Typhimurium 14028S (ATCC) transformed with *pDiGc* (Helaine et al., 2010) was used for infection according to infection protocol above. After lysing the RAW264.7 cells the lysates were washed once with sterile filtered PBS followed by analysis by flow cytometry (MACS, Miltenyi). Samples were gated for bacteria, GFP and dsRed (see "Fluorescent reporter assays" for settings). The % of bacteria that started to replicate was calculated by dividing the fraction of cells which lost RFP signal with at least 50% (S5 Fig) by $2^n$ where n is the number of generations the bacteria underwent during the experiment. The number of generations was determined by taking the $\log_2$ of the relative change in mean RFP value between the initial non-divided population and the dividing population (S5 Fig).

### Toxicity assay

Electro-competent SK4287, SK4288 and SK4289 were transformed with *pSK1913*, *pSK1914*, *pSK1915* or *pSK1918* by electroporation at 2.5 kV, 200 Ω, 25 μF and resuspended in 1 ml LB directly after. The transformed cells were incubated at 37°C for 1 h to recover. After recovery the transformed samples were split in two to which glucose or arabinose was added to a final concentration of 1% alongside with appropriate antibiotics for plasmid selection. Cultures were grown for 20 h at 30°C and plated for CFU counts on selective LA plates, supplemented with 1% glucose, by spotting.

### Quantitative RT-PCR

Bacteria for RNA extraction were grown over-night in LB or for 8h in RAW264.7 macrophages. LB cultures were diluted 1:1000 in LB broth and grown until $OD_{600}$ = 2.0 (ESP). RNA degradation and transcription were stopped by the addition of 2/5 v/v 95% ethanol and 5% phenol (pH 4.3) on ice for 30 min—2 h. Cells pellets were stored at -80°C until RNA extraction was performed using TRIzol (Invitrogen). DNA was removed using Turbo DNase I (Thermofisher) according to the manufacturer's instructions. Successful DNA removal was confirmed

by PCR using oligos SK411/SK412 (40 cycles), and RNA integrity vas validated on 2% agarose in 1xTEB, run at 80V. Total RNA was diluted to 5 ng/μl and RT-qPCR was performed using Brilliant Ultra-Fast SYBR Green QRT-PCR Master Mix (Agilent) with oligos SK362/SK363 (delivery) SK3/SK4 (*rhs-CT*$^{main}$) and SK5/SK6 (*rhs-CT*$^{orphan}$). The reactions were run on a StepOnePlus Real-Time PCR System (Applied biosystems). Relative gene expression was calculated using the $2^{-\Delta\Delta CT}$ method, normalizing the amount of target mRNA to expression of house-keeping gene *recA* amplified with oligos SK7/SK8 and wild type [57].

## Bioinformatic analyses

P2 promoter sequence from *S.* Typhimurium (CGGCTGCCGGGGCAGTATTTTGACGAC GAGACAGGGCTGCATTACAATCTGTTCAGATATTATGCACCGGAGTGTGGACGG TTTGTCAGTCAGGATCCGATCGGGCTG, 108bp) was used as the query for BLASTN search against the nt database (2019-04-29 release). 1147 hits with percent identity > = 80%, alignment length > = 95 bp and e-value < = 1e$^{-10}$ were kept, corresponding to 800 different genomes. Full-length genomes were then retrieved from NCBI, and sub-sequences matching the given promoter were obtained for further analysis, as well as an additional 138 bp downstream sequence directly after the promoter region. Promoter and the downstream sequences were aligned separately using MAFFT [58] v7.407 with the L-INS-i option, and the alignments were then merged and ordered after the similarity of the aligned promoter sequences.

## Supporting information

**S1 Fig. Overview of the plasmid encoding transcriptional reporters for the P1-P5 promoters.**
(PDF)

**S2 Fig. P2 promoter.** Overview of the **A)** P2$^{A}$ or **B)** P2$^{B}$ promoter sequences. -35 (yellow) and -10 (blue) regions were predicted using the softberry BPROM software. Transcription start sites (TSS´s) (bold and underlined) were determined using **C)** 5'RACE. Potential start codons are in green, RBS's in orange and in-frame stop codons in pink.
(PDF)

**S3 Fig. Toxicity of Rhs-CTmain and Rhs-CTorphan.** CFU counts from transformations of plasmids encoding arabinose inducible *rhs-CT*$^{main}$/*rhs-CT*$^{orphan}$ (ORF2) into NEB 5-α harboring no immunity or *rhsI*$^{main}$/*rhsI*$^{orphan}$ on a plasmids grown in either 1% glucose or 1% arabinose. n = 3, Error-bars are SEM. Statistical significance was determined using two-tailed students t-test where *** = P<0.005.
(PDF)

**S4 Fig. Detection of the translational activity of the P2$^{A-B}$ promoter driven toxin.** Single molecule fluorescence of strains with translational fusions of Rhs-CT$^{main}$ and Rhs-CT$^{orphan}$ ORF1's (CTG) to sYFP2. Strains were grown in M9-glucose to reduce background fluorescence. **A)** YFP fluorescence (au). **B)** Repeated experiment at higher laser voltage. Error-bars are SEM. Statistical significance was determined using two-tailed students t-test where *** = P<0.001 and ***** = P<0.00001.
(PDF)

**S5 Fig. Flow cytometric analyses to identify the dividing population of bacteria.** Representative graph of flowcytometric data used to identify the dividing population of bacteria after 16h of growth in RAW264.7 macrophages. The growing population is determined as the cells

where the dsRed fluorescent signal is decreased to levels below half of the mean fluorescent signal of the non-growing population.
(PDF)

**S6 Fig. Alignment of P2 promoter sequences and corresponding toxins.** Alignment showing P2 promoter-like sequences in bacterial genomes from NCBI. Downstream sequences illustrates different types of toxins the P2 promoters are found adjacent to. Homologous residues are shown in blue. Promoter sequences (-10, -35, TSS) and the conserved PxxxDPxGL motif are annotated above the sequences and demarcated with a black box.
(PDF)

**S1 Text.** Supplementary methods and Tables A-C.
(PDF)

**S1 Data. Raw data for all experiments included in this work.**
(XLSX)

# Acknowledgments

We thank Prune Leroy for the help with fluorescent microscopy and data analysis, Erik Holmqvist for the generous gift of the pEH167 plasmid and Fredrik Söderbom for the FS17 and FS354 oligos used in the study.

# Author Contributions

**Conceptualization:** Magnus Stårsta, Disa L. Hammarlöf, Marcus Wäneskog, Sanna Koskiniemi.

**Data curation:** Magnus Stårsta, Disa L. Hammarlöf, Arvid Heden Gynnå, Sanna Koskiniemi.

**Formal analysis:** Magnus Stårsta, Disa L. Hammarlöf, Marcus Wäneskog, Susan Schlegel, Feifei Xu, Arvid Heden Gynnå, Malin Borg, Sanna Koskiniemi.

**Funding acquisition:** Sanna Koskiniemi.

**Investigation:** Magnus Stårsta, Disa L. Hammarlöf, Marcus Wäneskog, Susan Schlegel, Feifei Xu, Malin Borg, Sten Herschend.

**Methodology:** Magnus Stårsta, Disa L. Hammarlöf, Marcus Wäneskog, Susan Schlegel, Feifei Xu, Malin Borg, Sanna Koskiniemi.

**Project administration:** Sanna Koskiniemi.

**Resources:** Sanna Koskiniemi.

**Supervision:** Disa L. Hammarlöf, Sanna Koskiniemi.

**Validation:** Magnus Stårsta, Disa L. Hammarlöf, Marcus Wäneskog, Malin Borg, Sten Herschend.

**Visualization:** Magnus Stårsta, Sanna Koskiniemi.

**Writing – original draft:** Magnus Stårsta, Disa L. Hammarlöf, Sanna Koskiniemi.

**Writing – review & editing:** Magnus Stårsta, Disa L. Hammarlöf, Marcus Wäneskog, Susan Schlegel, Feifei Xu, Arvid Heden Gynnå, Sanna Koskiniemi.

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
