## [Decision Letter · Decision Letter 0]

7 Sep 2019

Dear Dr Koskiniemi,

Thank you very much for submitting your Research Article entitled 'RHS-elements function as Type II Toxin-Antitoxin modules that regulate intra-macrophage replication of Salmonella Typhimurium' to PLOS Genetics. Your manuscript was fully evaluated at the editorial level and by four independent peer reviewers. The Editor and Reviewers found the findings of the study of significant interest and of potential importance to several fields, but also considered that certain issues need to be addressed and some of the findings consolidated before the paper is suitable for acceptance. Based on the reviews, we will not be able to accept this version of the manuscript, but we would be happy to consider a revised version. We cannot, of course, promise publication at that time.

Should you decide to submit a revised manuscript, your revisions should address the specific points made by each reviewer. We will also require a detailed list of your responses to the review comments and a description of the changes you have made in the manuscript. I would like to highlight several particular areas for your attention: Reviewer 1 raises pertinent issues about further validation of the toxin-antitoxin function of the RhsCT-RhsI pairs, and several reviewers agree that the toxicity assays in Figure 3 are inadequate and a quantitative assay is required (such as viable count determination during liquid growth). Several reviewers highlight that differences in the behaviour of the two RhsCT-RhsI pairs should be more explicitly considered and it is important that you address the question of why the orphan RhsCT-RhsI pair seems to be almost non-toxic in a heterologous expression assay yet has the stronger phenotype in macrophages. Reviewer 5 details some improvements required for the experiments and presentation of Figure 5, including complementation of the mutants, whilst several Reviewers have some questions about the relevance of the data currently presented in Figure 4.

If you decide to revise the manuscript for further consideration at PLOS Genetics, please aim to resubmit within the next 60 days, unless it will take extra time to address the concerns of the reviewers, in which case we would appreciate an expected resubmission date by email to plosgenetics@plos.org.

[LINK]

We are sorry that we cannot be more positive about your manuscript at this stage. Please do not hesitate to contact us if you have any concerns or questions.

Yours sincerely,

Sarah J Coulthurst, PhD

Guest Editor

PLOS Genetics

Lotte Søgaard-Andersen

Section Editor: Prokaryotic Genetics

PLOS Genetics

Reviewer's Responses to Questions

**Comments to the Authors:**

Reviewer #1: The manuscript from Hammarlöf et al. describes functional analysis of Rhs toxin systems from Salmonella Typhimurium. The Rhs locus contains two sets of toxin and inhibitor-coding ORFs. Internal promoters are shown to allow expression of these two pairs, suggesting their activity as type II toxin-antitoxin systems. Deletion analysis of Rhs locus demonstrated that these toxin pairs reduce growth in macrophages.

This is an interesting study which blends the fields of protein toxin secretion and internal toxin-antitoxin systems. On the whole, the data are well presented. There are, however, multiple unresolved questions and the current claims require further support before these two systems can truly be considered toxin-antitoxin systems.

Major points:

1) The assays of Figures 3 and S3 are poor and do not clearly demonstrate the extent of toxicity nor reproducibility. As the authors have the strains at hand, I would strongly recommend that the assays are repeated as full viable count assays using vector only controls where appropriate.

2) To be considered type II toxin-antitoxin systems, the authors should demonstrate (i) an interaction between the two components and (ii) whether or not the antitoxin and/or TA complex perform transcriptional autoregulation, as per other TA pairs. Perhaps the first point has been published in previous Rhs literature?

3) The toxicity assays demonstrate that rhs-orphan is not toxic. Yet, the Rhs-orphan mutant has the biggest impact in the macrophage assays, allowing the mutants to replicate. This implies Rhs-orphan is in fact toxic under certain conditions perhaps? Could this be addressed in the discussion?

4) Can the authors speculate further about toxin targets, from modelling or database searches?

5) Can the authors discuss in further detail regarding the lack of phenotype for rhs-complete in figure 5C?

Minor points:

1) Line 93, missing word

2) Line 128 – please define “rhs gene” more clearly for non-expert readers., ie the full ORF from RHS-delivery to RhsI-main.

3) Lines 137-139, re-order so ESP comes first, then MAC, so it matches the figures.

4) Line 211 missing superscript

5) Figures 5D and 5C have been swapped over in the text/figure, they do not match (lines 227-233.

6) Figure 5C (as on figure), the y axis is labelled oddly – what does it mean?

7) Figs 5B-D, lighten shading of bars to make error bars more visible.

Reviewer #2: Review

Manuscript ID: PGENETICS-D-19-01316

Title: RHS-elements function as Type II Toxin-Antitoxin modules that regulate intra-macrophage replication of Salmonella Typhimurium

Corresponding Author: Koskiniemi

Summary

This work describes the expression and the role of Rhs toxins, namely Rhs-CTmain and Rhs-CTorphan, in Salmonella Typhimurium during macrophage infection. These Rhs toxins are in the type II toxin-antitoxin (TA) system and also exhibit contact-dependent growth inhibition (CDI). The authors first identify the internal promoters of the rhs genes and then demonstrate that the two toxins can be expressed from those regions, which is one of the characteristic traits of the type II TA modules.

Then, they confirm that the internal transcription yields the toxins by expressing them in Escherichia coli under arabinose-inducible promoter. The bacteria exhibit a growth defect on LB agar plates with L-arabinose only when the region that encodes the cognate immunity protein is not included. The authors also measure the levels of proteins produced from the internal transcripts by inserting YFP downstream of the start codon and using fluorescence microscopy to determine the YFP signal as a read-out. In addition, the authors demonstrate that the cognate immunity proteins are susceptible to degradation by a stress-induced protease, Lon and propose the mechanism how Rhs toxins become active inside macropahges. Finally, by using genetic knockouts and flow cytometry, their data indicate that the main role of Rhs toxins is to arrest growth of Salmonella upon entry into macrophages.

Overall, these findings explain the expression data from the Hinton Lab (Srikumar et al., PLoS Pathog., 2015) why rhs genes are up-regulated when bacteria are inside macrophages. The experiments are well-designed. Although the finding that Rhs toxins seem to be important in intracellular replication, many of the other findings are not all that new and surprising.

Major Comments:

1. In Figure 3 under the rhs-CTmain ORF1(CTG) column (left column), why does the E. coli strain without any plasmid not grow on the plate in the presence of L-arabinose?

2. The authors propose that the cognate immunity proteins of Rhs toxins are degraded by Lon, which is also up-regulated inside macrophages. If that is the case, how do the bacterial cells that express Rhs toxins remain viable once the cognate immunity proteins are degraded? What happens in individual Salmonella cells with regard to Lon expression and degradation of immunity proteins to Rhs? Doesn’t Lon degrade many anti-toxins?

3. Following up from question 2, why does Salmonella have both Rhs-CTmain and Rhs-CTorphan if they are mainly used to control the intracellular growth of the pathogen?

4. Rhs toxins seem to have multiple roles. The previous work done by Koskiniemi (PLOS genet., 2014) suggest that Rhs toxins are used for a clonal selection. Although it is already stated in the discussion, it might be worth having a small paragraph to discuss the two studies more clearly somewhere.

5. Could the authors speculate why these Rhs toxins are also expressed in some other conditions like in early stationary phase (ESP)?

6. Are the Rhs toxins mediating contact-dependent killing of sister cells within the macrophage?

Minor Comments:

1. From line 225 onwards, please check if the Figure 5C and 5D are swapped. The description in texts is not matched with the data present in each figure.

2. Line 422, should the word “YFP” be replaced with GFP or FITC? Or did the authors actually use the setting for YFP to gate the GFP signal from the pDiGc plasmid?

Reviewer #3: In this manuscript he authors demonstrate that Rhs elements in S. Typhimurium function as toxin-antitoxin systems and contribute to growth regulation in macrophages. They present a possible model for how the Rhs do this by demonstrating that their activity is regulated by a protease known to be expressed during intracellular growth. The experimental plan and conclusions are overall very solid. Below are some comments:

For Figure 4, I'm not sure I understand what is being shown. For one, there is a lot more than 6-fold more expression with ATG instead of CTG. I'm not convinced that you're actually measuring the protein production level of the native gene using this construction. If you wanted to actually measure the protein production level, you should make an inactivating mutant in the toxin then use a quantitative western or something like that. You know that the promoter is active. Regardless of how much is produced, you know that enough toxin is made to restrict growth of E. coli. You aren't accurately quantifying protein production levels, so this figure doesn't add anything.

For the text discussion of figure 5, I was really confused... I think the references to figure 5c and 5d are switched? Anyway, the biggest difference between mutants in 5b is between the 3rd and 4th columns. The authors seem to be suggesting that this difference is explainable by differences in different delays in replication starting, but the bar graphs don't really show that – the ones with more survival have a longer delay before starting replication. Maybe I just don't understand how the measurements are being made. Maybe the authors could reword this paragraph to make it more clear.

I couldn't help but notice that the authors avoided commenting on differences between Main and Orphan in their data. For example, in fig 6, it seems like lon has a much greater impact on Orphan than the main cluster. Similarly, in Figure 4, there are some differences between Main and and Orphan. Can anything be concluded about the roles of the two CTs?

Reviewer #4: Review uploaded

**Have all data underlying the figures and results presented in the manuscript been provided?**

Reviewer #1: Yes

Reviewer #2: Yes

Reviewer #3: Yes

Reviewer #4: Yes

PLOS authors have the option to publish the peer review history of their article (what does this mean?). If published, this will include your full peer review and any attached files.

Reviewer #1: No

Reviewer #2: No

Reviewer #3: No

Reviewer #4: No

---

## [Decision Letter · Decision Letter 1]

12 Jan 2020

Dear Dr Koskiniemi, 

We are pleased to inform you that your manuscript entitled "RHS-elements function as Type II Toxin-Antitoxin modules that regulate intra-macrophage replication of Salmonella Typhimurium" has been editorially accepted for publication in PLOS Genetics. Congratulations!

As you will see from the comments below, all of the Reviewers were supportive of your revised manuscript being accepted for publication, but they did note one or two very minor typographical errors. Please ensure these are corrected during the formatting checks mentioned below.

Yours sincerely,

Sarah J Coulthurst, PhD

Guest Editor

PLOS Genetics

Lotte Søgaard-Andersen

Section Editor: Prokaryotic Genetics

PLOS Genetics

Comments from the reviewers (if applicable):

Reviewer's Responses to Questions

**Comments to the Authors:**

Reviewer #1: This revised manuscript by Hammarlof et al satisfactorily answers all queries raised at the initial review stage. The findings are now clearer and the conclusions are better supported. Caveats are well discussed and on the whole the manuscript is much stronger. I have only one minor change to make - there is a typo in the y-axis label for 5C. This article should be of wide interest to microbiologists.

Reviewer #2: The authors have addressed my concerns.

Reviewer #3: The authors have addressed all of my concerns.

The redrawn figure 3 is now much more clear/convincing, but it looks like the figure title in the figure legend is duplicated.

Reviewer #4: The authors have successfully addressed all my points, together with those of other reviewers. This is now ready for publication.

**Have all data underlying the figures and results presented in the manuscript been provided?**

Reviewer #1: Yes

Reviewer #2: Yes

Reviewer #3: Yes

Reviewer #4: Yes

PLOS authors have the option to publish the peer review history of their article (what does this mean?). If published, this will include your full peer review and any attached files.

Reviewer #1: No

Reviewer #2: No

Reviewer #3: No

Reviewer #4: No

**Data Deposition**

http://datadryad.org/submit?journalID=pgenetics&manu=PGENETICS-D-19-01316R1

**Press Queries**

---

## [Editor Report · Acceptance letter]

4 Feb 2020

PGENETICS-D-19-01316R1 

RHS-elements function as Type II Toxin-Antitoxin modules that regulate intra-macrophage replication of Salmonella Typhimurium 

Dear Dr Koskiniemi, 

We are pleased to inform you that your manuscript entitled "RHS-elements function as Type II Toxin-Antitoxin modules that regulate intra-macrophage replication of Salmonella Typhimurium" has been formally accepted for publication in PLOS Genetics! Your manuscript is now with our production department and you will be notified of the publication date in due course.

With kind regards,

Matt Lyles

PLOS Genetics

On behalf of:
